# A spontaneous termination mechanism of RNA polymerase V shapes the DNA methylation landscape in plants

Guohui Xie [ID] [1,8], Xuan Du [ID] [2,8], Yifang Tan [1,8], Yuxing Zhou [ID] [3,8], Cheng Chi [4,8], Sixian Zhou [1], Colette L Picard [ID] [3], Songge Chai [ID] [1], Lei Wu [ID] [1], Danling Zhu [ID] [1], Jun Zhao [ID] [4], Yan Xue [ID] [4], Sisi Li [ID] [5], Steven E Jacobsen [ID] [3,6 ✉], Zhe Wu [ID] [1 ✉] & Jiamu Du [ID] [1,7 ✉]

## Abstract

DNA methylation plays critical roles in eukaryotic gene silencing, genome imprinting, viral defense, and suppression of transposable elements. In plants, RNA Polymerase V (Pol V)-generated non-coding RNA guides DNA methylation through the RNA-directed DNA methylation (RdDM) pathway; however, how these RNAs are selected is unknown. Here, we show that the 3′-ends of Pol V transcripts are enriched at A-rich template DNA (A-rich-DNA$_T$). Arabidopsis RdDM regions possess AT-rich boundaries genome-wide, suggesting that Pol V likely terminates at A-rich-DNA$_T$, which subsequently defines the DNA methylation landscape in plants. A-rich-DNA$_T$ successfully stops Pol V transcription in vitro. Structural snapshots of Pol V transcribing A-rich-DNA$_T$ show that accumulation of unstable rU:dA pairs in the RNA-DNA hybrid promotes transcription bubble collapse and spontaneous transcription termination. These findings identify an intrinsic Pol V termination signal that shapes genomic DNA methylation patterning in plants and reveals a common mechanism for spontaneous transcription termination.

**Keywords** RNA-directed DNA Methylation; RNA Polymerase V; Transcription; Spontaneous Termination; Structural Biology
**Subject Categories** Chromatin, Transcription & Genomics; Plant Biology; Structural Biology

## Introduction

Genetic information encoded in DNA is transcribed into RNA by the DNA-dependent RNA polymerases (RNAPs) in three steps: initiation, elongation, and termination (Girbig et al, 2022a). During transcription termination, RNAPs need to perceive the termination signal, stop the transcription elongation, and release the RNA transcripts, which is a multiple-step complex process. Termination is pivotal for complete and successful transcription and is mediated either by DNA sequences (spontaneous termination) or by protein cofactors (Girbig et al, 2022a; Richardson, 1993). In bacteria, a hairpin structure preceding a poly(U) transcript mediates spontaneous termination, while the Rho ATPase mediates cofactor-based termination (Hao et al, 2021; Molodtsov et al, 2023; Ray-Soni et al, 2016; Said et al, 2021; You et al, 2023). In archaea, the ribonuclease FttA cleaves and translocates RNA for cofactor-based termination (Sanders et al, 2020; You et al, 2024). In eukaryotes, termination is more complex. Understanding of the termination of RNAP II (Pol II), one of the most studied eukaryotic RNAPs, remains controversial with two predominant models: the torpedo model mediated by the exonuclease Rat1-Rai1 to cleave RNA transcripts for cofactor-based termination, and the allosteric model mediated by conformational change of Pol II (Kim et al, 2004; West et al, 2004; Yanagisawa et al, 2024; Zeng et al, 2024; Zhang et al, 2015a). However, termination by Rat1-Rai1 is less efficient in vitro and seems unable to fully support efficient Pol II termination in vivo, while the allosteric conformational change of Pol II remains unclear (Dengl and Cramer, 2009; Han et al, 2023; Park et al, 2015). Although the DNA sequence is undoubtedly crucial for controlling the entire transcription process, neither the torpedo nor allosteric model takes DNA sequences into account. Thus, the impact of DNA sequences on Pol II termination is uncertain and underestimated. While it has long been reported that T-rich non-template DNA (T-rich-DNA$_{NT}$) can induce Pol II termination in

[1]Shenzhen Key Laboratory of Plant Genetic Engineering and Molecular Design, Institute of Plant and Food Science, Department of Biology, School of Life Sciences, Southern University of Science and Technology, Shenzhen, China. [2]Guangdong Provincial Key Laboratory for Plant Epigenetics, Shenzhen Key Laboratory of High-Efficiency Utilization of Light in Plants, College of Life Sciences and Oceanography, Shenzhen University, Shenzhen, China. [3]Department of Molecular, Cell and Developmental Biology, University of California at Los Angeles, Los Angeles, CA, USA. [4]Shandong Laboratory of Advanced Agricultural Sciences at Weifang, Peking University Institute of Advanced Agricultural Sciences, Weifang, China. [5]International Cancer Center, Guangdong Key Laboratory of Genome Instability and Human Disease Prevention, Department of Biochemistry and Molecular Biology, Shenzhen University Medical School, Shenzhen, China. [6]Howard Hughes Medical Institute, University of California at Los Angeles, Los Angeles, CA, USA. [7]Institute for Biological Electron Microscopy, Southern University of Science and Technology, Shenzhen, China. [8]These authors contributed equally: Guohui Xie, Xuan Du, Yifang Tan, Yuxing Zhou, Cheng Chi. ✉E-mail: jacobsen@ucla.edu; wuz@sustech.edu.cn; dujm@sustech.edu.cn

some animal genes without the hairpin structure, it was only recently identified as a general Pol II termination signature genome-wide in vivo, implying a plausible eukaryotic spontaneous termination mechanism different from bacteria (Bentley and Groudine, 1988; Davidson et al, 2024; Han et al, 2023; Reines et al, 1987).

In plants, in addition to the canonical Pol I, Pol II, and Pol III that are conserved in all eukaryotes, two atypical RNAPs, Pol IV and Pol V, have evolved and play pivotal roles in the plant-specific RNA-directed DNA methylation (RdDM) pathway to mediate the de novo establishment of DNA methylation (Matzke and Mosher, 2014; Ream et al, 2009; Roeder and Rutter, 1969; Xie et al, 2025). In RdDM, Pol IV transcripts are successively processed by RNA-DEPENDENT RNA POLYMERASE 2 and DICER-LIKE 3 to produce 24-nt small interfering RNA (siRNA) to load into ARGONAUTE 4 (AGO4) (Herr et al, 2005; Huang et al, 2021; Onodera et al, 2005; Pontier et al, 2005; Wang et al, 2021; Xie et al, 2005; Zilberman et al, 2004). Meanwhile, Pol V transcribes scaffold long non-coding RNA to recruit the AGO4-siRNA complex to direct DOMAINS REARRANGED METHYLTRANSFERASE 2 (DRM2) for locus-specific DNA methylation (Cao and Jacobsen, 2002; Huang et al, 2009; Wierzbicki et al, 2008; Wierzbicki et al, 2009; Zhong et al, 2014). Given the scaffold function of Pol V in RdDM to recruit DRM2, the transcription initiation and termination sites of Pol V should, in principle, mirror the boundaries of RdDM on chromatin. Biochemically, Pol V features a low elongation rate and enhanced backtracking, a backward movement of the RNAP along the DNA to trigger the 3'–5' RNA cleavage for the proofreading and regulations, underlying its chromatin retention mechanism to support its scaffold function (Xie et al, 2024; Xie et al, 2023; Zhang et al, 2023). However, the site-specificities and molecular mechanisms of the initiation and termination of Pol V, which are keys to understanding the chromatin landscape of RdDM, remain unclear.

# Results

## Combined NET-sequencing and RIP-sequencing reveal Pol V termination signals

Despite being identified for more than 20 years (Herr et al, 2005; Onodera et al, 2005; Pontier et al, 2005), low-level endogenous accumulation still poses challenges for high-quality sequencing of Pol V native transcripts. RNA immunoprecipitation sequencing (RIP-seq) and global nuclear run-on sequencing have been used to investigate the 5'-end and global features of Arabidopsis Pol V transcripts (Bohmdorfer et al, 2016; Liu et al, 2018), but the 3'-ends are not well captured with these methods. To precisely characterize the 3'-end features of Pol V transcripts, we subjected our large-scale and high-purity cauliflower (*Brassica oleracea* var. *botrytis*) Pol V (Xie et al, 2023) to both native elongating transcript sequencing (NET-seq) and RIP-seq for mapping Pol V-protected fragments and Pol V-bound transcripts, respectively (Appendix Fig. S1). The two biological replicates in our NET-seq showed good reproducibility, yielding more than 20 million unique mapped reads (Appendix Fig. S2A,B). As expected, most of the Pol V NET-seq reads (~69%) were mapped to intergenic regions (Bohmdorfer et al, 2016) (Appendix Fig. S2C). Our de novo assembling of NET-seq

reads produced 8197 putative non-overlapping Pol V transcripts (Appendix Fig. S2D). Meanwhile, 6780 Pol V transcripts were assembled from the RIP-seq data, which are highly correlated to NET-seq (Appendix Fig. S2D–F), supporting the reliability of both experiments. In principle, the 3'-ends of the RIP-seq assembled transcripts should approximate Pol V termination sites. Thus, we defined −50-bp to +200-bp around the 3'-end of each RIP-seq transcript as the termination region (Fig. 1A). Given that the precise 3'-end information is kept in NET-seq data, we then analyzed the NET-seq transcripts whose 3'-ends were located within the termination region (Fig. 1A). Among the 8197 NET-seq transcripts, 2472 have their 3'-ends located in the terminating regions. The remaining 5725 NET-seq transcripts were mostly either located within the gene bodies of RIP-seq-assembled transcripts (1024 transcripts) or exhibited RIP-seq read signals at their 3'-ends but were insufficient for transcript assembly under the stringent parameters of our pipeline (2953 transcripts). To ensure stringency, subsequent analyses were restricted to the 2472 transcripts whose 3'-end was jointly supported by both datasets. Sharp Pol V peaks were observed at the 3'-end of the 2472 NET-seq transcripts by metagene analysis and at individual loci (Fig. 1B,C), suggesting Pol V arrests at these positions. Notably, the 3'-end enrichment features were also robustly observed when plotting all 8,197 NET-seq transcripts (Appendix Fig. S2G). In addition, to exclude potential interference from transcript abundance on the metagene analysis, we also plotted the distribution of the positions with the highest Pol V NET-seq signal along each transcript. The results further support that the accumulation at the transcript end is a prominent feature of Pol V transcription (Appendix Fig. S2H). Intriguingly, for the 2472 NET-seq transcripts, their 3'-ends on the non-template strand exhibit a pronounced T-rich feature compared with the gene body (Appendix Fig. S2I). Further motif enrichment analysis at the 3'-ends revealed enrichment of poly(U) sequences (Fig. 1D). Specifically, motifs consisting of consecutive T on the non-template strand were much more frequent at the 3'-ends of NET-seq transcripts than either other nucleotides at the same region or consecutive T within the gene body. (Fig. 1E; Appendix Fig. S2J). Thus, we show that T-rich-$DNA_{NT}$ is highly enriched at Pol V termination sites and correlated with Pol V arrest, implying a potential DNA sequence-dependent termination mechanism of Pol V.

## Arabidopsis RdDM has a T-rich-$DNA_{NT}$ boundary

Pol V functions as a scaffold to guide RdDM via interactions between its transcripts and 24-nt siRNA, which ultimately targets the DRM2 DNA methyltransferase to methylate DNA (Xie et al, 2025). We therefore hypothesized that if T-rich-$DNA_{NT}$ terminates Pol V transcription in vivo, the boundaries of RdDM regions, which are well defined in Arabidopsis, should be enriched for this sequence. We reanalyzed published Arabidopsis NRPE1 (the largest subunit of Pol V) chromatin immunoprecipitation followed by sequencing (ChIP-seq) data and found that NRPE1 ChIP-seq peak edges are highly AT-rich (Liu et al, 2018) (Fig. 2A), particularly over the regions most highly bound by NRPE1 (Appendix Fig. S3A), suggesting that runs of Ts may be more common at the boundaries of Pol V-transcribed regions. To investigate the DNA sequence-based Pol V termination signal in Arabidopsis further, we obtained a list of 4502 published transcript end sites (TES) from Arabidopsis

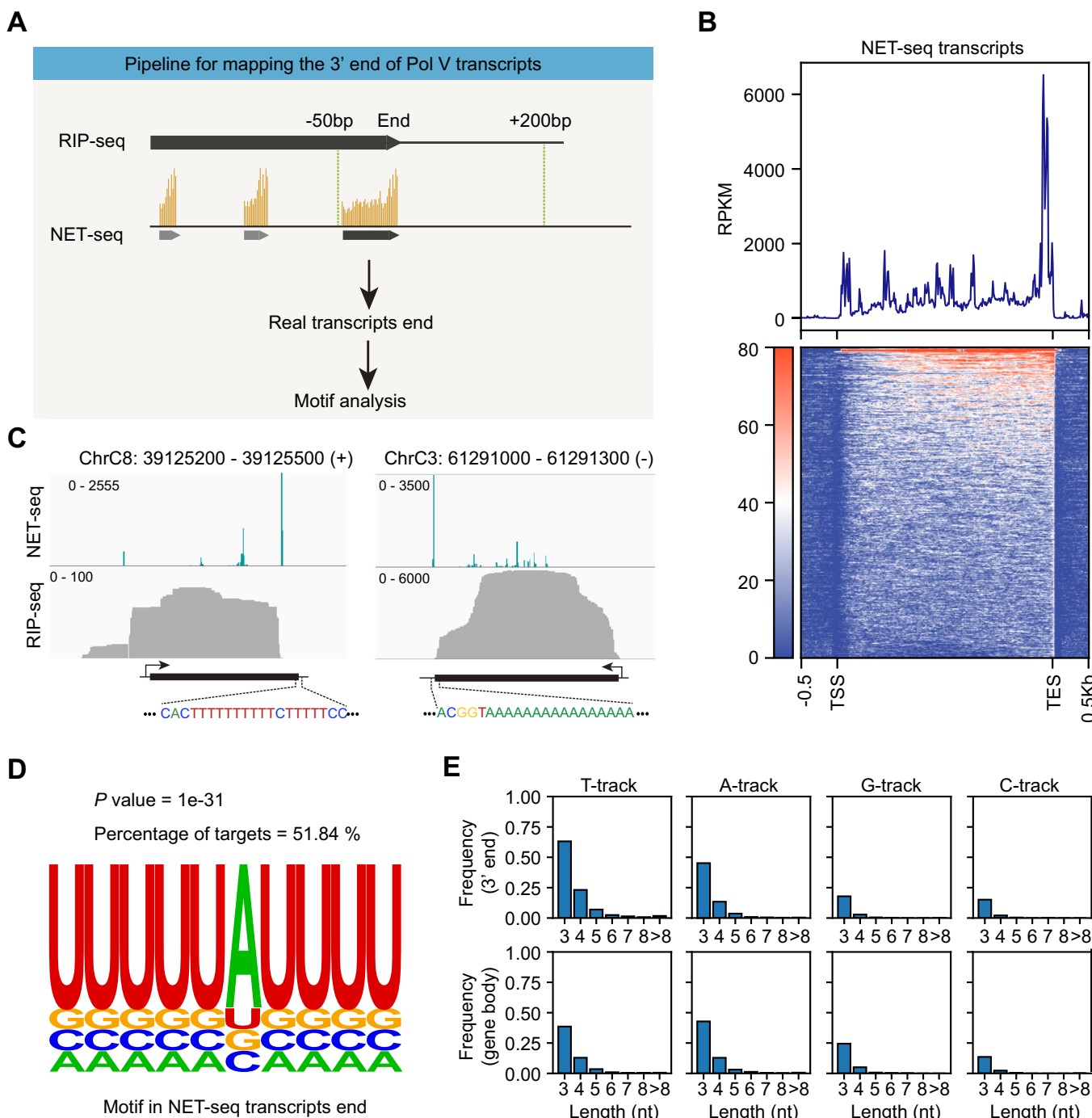

**Figure 1. Pol V NET-seq and RIP-seq analysis reveal its termination features.**

(A) Schematic illustration of the pipeline used for mapping the 3'-end of Pol V transcripts. (B) Metagene plot showing the Pol V positioning along the NET-seq data assembled transcripts with their 3'-ends located in the terminating regions as illustrated in (A). (C) Genome browser tracks showing the normalized counts from Pol V NET-seq data (1-nt resolution) and reads from Pol V RIP-seq data of two example Pol V transcripts. Transcripts are indicated by black boxes. Arrows indicate the direction of transcription. Bases in the enlargement area are 21-nt sequences of $DNA_{NT}$ (left) and $DNA_T$ (right) centered on the transcript end sites. (D) RNA motif enrichment within -20-nt to +20-nt of the transcripts' end of the 2472 NET-seq transcripts. (E) The frequencies of motifs consisting of different lengths of consecutive T/A/G/C at transcript 3' ends (−20-nt to +20-nt of non-template strand) and in random windows (40-nt) within Pol V transcript bodies.

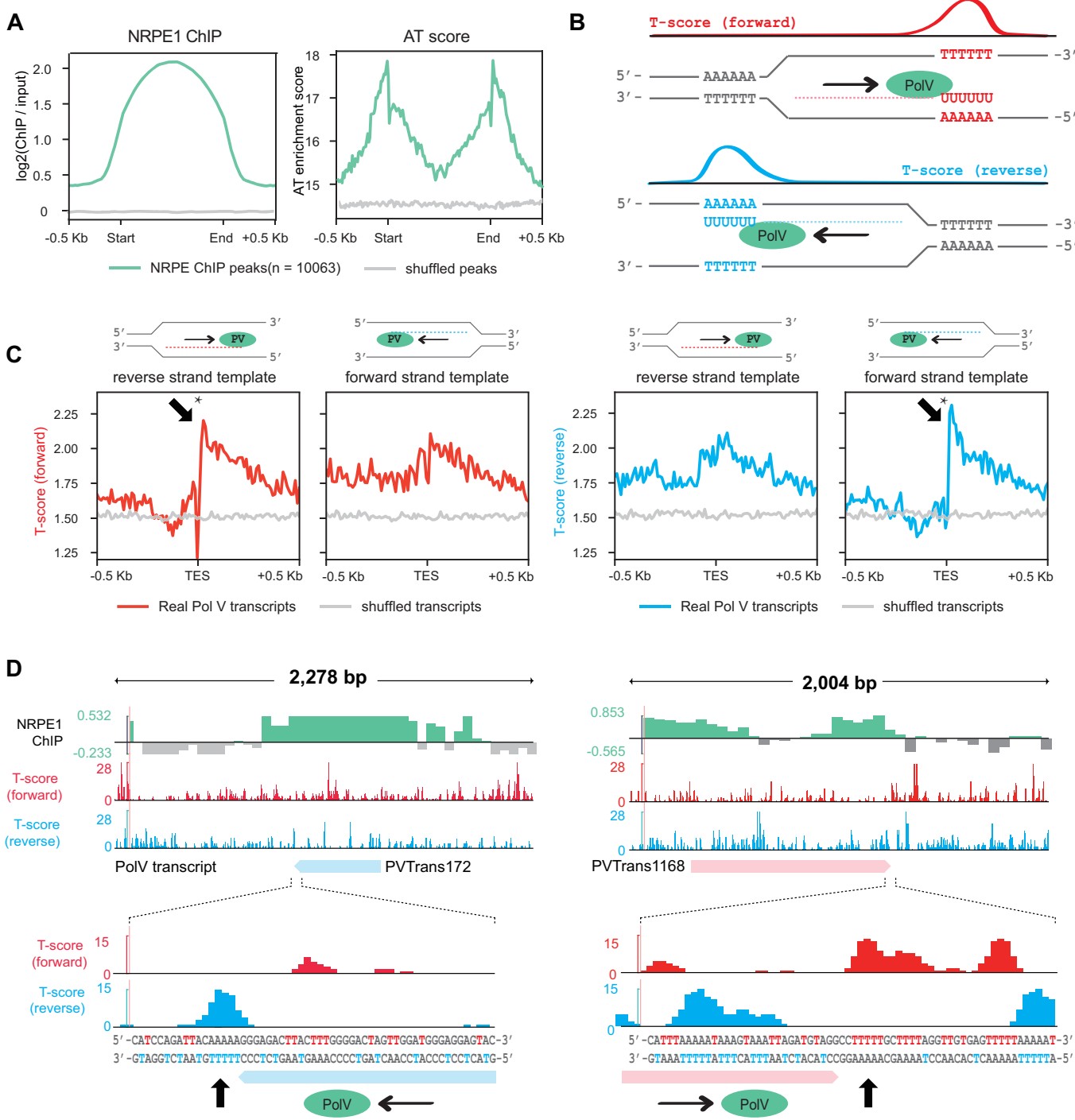

**Figure 2. Arabidopsis Pol V transcript ends are enriched for T-rich non-template DNA.**

(A) Metaplots of average NRPE1 ChIP-seq signal over NRPE1 ChIP-seq peaks, alongside "AT-score", which scores the AT-richness of the regions. (B) Diagram of T-score on forward strand (top) vs. reverse strand (bottom). T-score increases the more consecutive Ts are on the indicated strand (forward or reverse). T-score (forward) will be high when a series of consecutive Ts are encountered on the forward strand, while T-score (reverse) will be high when consecutive Ts are present on the reverse strand. The diagram also shows Pol V transcribing using the reverse strand as template (top) or the forward strand as template (bottom). (C) Metaplots over Arabidopsis Pol V transcript end sites (TES) of T-score of either forward (left) or reverse (right) strand. Note that all metaplots are oriented relative to transcription direction, so that left part of metaplot corresponds to sequences before termination, and right after termination. Pol V transcripts using the forward and reverse strands as template were plotted separately. *P value < 2.2e-16, Welch Two-Sample t test (50-bp regions upstream TES compared to 50-bp regions downstream TES). (D) Example Pol V RIP-seq transcripts and their termination sites, showing distribution of T-scores (forward/reverse) around these sites, as well as NRPE1 ChIP-seq signal.

Pol V RIP-seq (Bohmdorfer et al, 2016). Of the 4502 published Pol V transcripts, 4002 (88.9%) overlapped with the NRPE1 ChIP-seq peaks, demonstrating that the published RIP-seq transcriptome is highly consistent with our reanalyzed ChIP-seq dataset. We calculated a "T-score" genome-wide for the forward and reverse strands separately, which identifies regions where the Arabidopsis genome sequence contains runs of successive Ts (Fig. 2B; Appendix S3B). We found that for Pol V transcripts using the forward strand as template, the reverse (i.e., non-template) strand T-score was low across the transcript body, but sharply increased immediately after the TES (Fig. 2C,D; Appendix S3C,D). Similarly, for Pol V transcripts using the reverse strand as template, the forward (i.e., non-template) strand T-score was also low across the transcript body, but strongly increased immediately after the TES (Fig. 2C,D; Appendix S3C,D). By contrast, average T-scores on the template strand of Pol V transcripts did not vary substantially around the TES (Fig. 2C,D; Appendix S3D), although they were generally elevated, consistent with the edges of RdDM regions being generally AT-rich (Fig. 2A). Together, these data suggest that T-rich-$DNA_{NT}$ can also function as a termination signal for Pol V in Arabidopsis that helps define RdDM boundaries, and may be a generally conserved feature of Pol V termination. Interestingly, we also observed high T-scores at the TSS of Pol V transcripts (Appendix Fig. S3D). Plant genes are known to contain runs of Ts near the TSS, which are thought to contribute to transcription initiation by Pol II as well as destabilization of nucleosomes (Zhang et al, 2022; Zhang et al, 2015b). As a common feature required for the separation of the DNA duplex upon transcription initiation, we speculated that the T enrichment at Pol V TSS sites serves a similar purpose. Together, these results support the hypothesis that successive runs of Ts may serve as termination signals for Pol V transcripts in vivo.

## A-rich-$DNA_T$ arrests Pol V transcription in vitro

To investigate whether T-rich-$DNA_{NT}$ induces Pol V termination directly or indirectly, we performed an in vitro transcription assay with a series of designed transcription bubbles as substrates (Fig. 3A; Appendix Table S1). Compared to regular transcription elongation with a reported common substrate (Xie et al, 2023), a designed transcription bubble substrate with a poly(A) at the transcribing region of $DNA_T$ ($DNA_T$-TR) and a regular sequence at the unpaired region of $DNA_{NT}$ ($DNA_{NT}$-UPR) almost fully eliminates transcription elongation of Pol V (Fig. 3B, scaffold 2), implying a failure of transcription initiation or a possible transcription pausing or termination. Moreover, it is worth noting that scaffold 2 in the elongation assay has abnormally strong RNA degradations (Fig. 3B). RNAPs always backtrack and subsequently cleave the RNA from the 3'-end once the forward movement is blocked, for example, by the misincorporation of improper nucleotides (Nudler, 2012). Therefore, we considered that the abnormal RNA degradation of scaffold 2 in our assay plausibly implies a less-efficient forward elongation and/or an immediate termination, induced backtracking, and the subsequent backtracking-induced RNA cleavage. Further, bubbles with a regular sequence at $DNA_T$-TR but poly(T) or poly(A) at the $DNA_{NT}$-UPR restored elongation of Pol V (Fig. 3B, scaffolds 3 and 4), suggesting that the sequence of $DNA_{NT}$ is dispensable for stopping the Pol V transcription.

Moreover, we placed poly(A) or poly(T) sequences in the downstream paired dsDNA region of $DNA_T$ flanked by a 2-nt unpaired sequence to allow Pol V to initiate transcription. After successfully transcribing the 2-nt unpaired region, poly(A) but not poly(T) in the $DNA_T$ stopped transcription (Fig. 3C, scaffolds 5 and 6). Both scaffolds 5 and 6 have a product accumulation at the 2-nt elongation product (Fig. 3C), corresponding to the 2-nt linker between the 3'-end of the RNA primer and the poly(T) or poly(A) sequence (Fig. 3A) and consistent with our previous report that the Pol V may arrest at the downstream DNA branching site upon elongation (Xie et al, 2023). However, the scaffold 5 stopped immediately without clear further elongation (Fig. 3C), suggesting a transcription pausing or termination upon transcribing poly(A) template. In contrast, the scaffold 6 yields longer transcripts beyond the accumulated 2-nt product. However, the transcription elongation efficiency for scaffold 6 is lower than that of scaffold 1 of a regular sequence (Fig. 3C). Considering that the rU:dA pair is less stable than the rA:dT pair and the A:T pair is less stable than the G:C pair, we consider that the less stable pairing between the RNA transcript and the $DNA_T$ may plausibly prevent elongation and cause the transcription pausing and/or termination (see "Discussion" below).

Further, we asked whether A-rich-$DNA_T$-based Pol V transcription stopping is induced by the weak A:T(U) pairs or is strictly poly(A)-dependent. Compared to poly(A)-$DNA_T$-TR, the poly(T)-$DNA_T$-TR did not induce transcription stopping (Fig. 3D, scaffolds 2 and 7), suggesting a strict requirement of A-rich-$DNA_T$. Therefore, we define the termination sequence of Pol V as the A-rich-$DNA_T$, but no longer use the T-rich-$DNA_{NT}$ in the following discussion. We further tested positional effects and found that both the TATATATA and TTTTAAAA sequences of $DNA_T$ failed to yield sufficient elongation product, suggesting a failure of Pol V transcription initiation, or an immediate transcription stop or termination (Fig. 3D, scaffolds 8 and 9). Similarly, a CA repeat or a GA repeat sequence of $DNA_T$ yields lower or nearly no elongation product, too (Appendix Fig. S4A,B). Overall, we demonstrate that sufficient As in $DNA_T$ is a key determinant to inducing Pol V transcription pausing or termination in vitro.

## Molecular basis of Pol V termination

To investigate the molecular basis of A-rich-$DNA_T$-induced Pol V termination or pausing, we carried out structural studies to take snapshots of Pol V transcribing an A-rich-$DNA_T$ site, using cryogenic electron microscopy (cryo-EM) (Fig. 4; Appendix Figs. S5–14; Appendix Tables S1 and 2). A UTP analog, uridine 5'-[(β,γ)-imido]-triphosphate (UMPPNP), was incorporated to mimic the upcoming substrate UTP and thereby stop the reaction for fixing Pol V transcribing complexes at each step. When the A-rich-$DNA_T$ sequence is yet-to-be transcribed at the downstream dsDNA region, the Pol V transcription complex adopts a normal elongation conformation almost identical to that observed in our previous study (Xie et al, 2023) (Fig. 4B,C). This does not induce a significant conformational change of either the Pol V or the transcription bubble (Fig. 4B,C; Appendix S14A). Consistently, scaffold 5 can transcribe 2-nt in our biochemical assay (Fig. 3B).

To mimic the process of Pol V continuously transcribing over an A-rich-$DNA_T$ site, we stepwise fed consecutive As into the $DNA_T$-TR from downstream to upstream to yield the corresponding

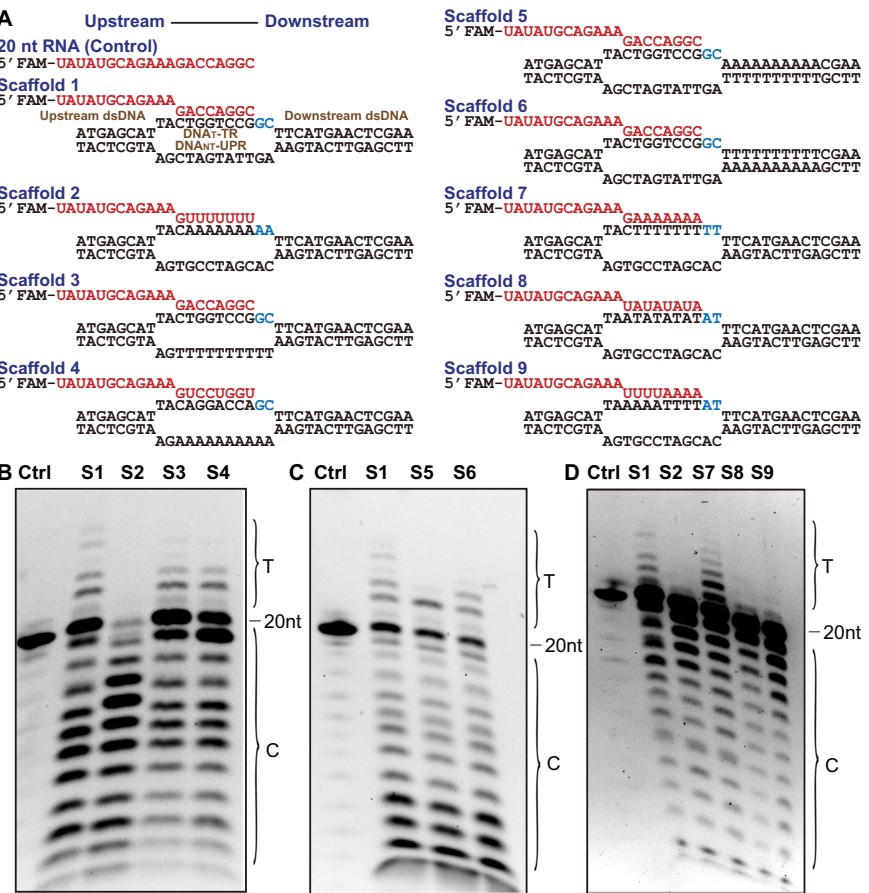

**Figure 3. In vitro Pol V transcription assay.**

(A) The designed nucleic acid scaffolds for in vitro assays. The RNA is labeled by 5'-fluorescein amidite (FAM) and highlighted in red. The structural elements of the designed scaffold are labeled. $DNA_T$-TR, transcribing region of the template DNA; $DNA_{NT}$-UPR, unpaired region of the non-template DNA. (B) Transcription assay shows that A-rich-$DNA_T$ can stop Pol V transcription (S2), while the sequence of $DNA_{NT}$ is dispensable for this effect (S3 and S4). Ctrl control, S1 scaffold 1, T transcription elongation product, C cleavage product. (C) Transcription assay shows that Pol V stops upon transcribing over A-rich-$DNA_T$ (S5), and elongates upon T-rich-$DNA_T$ (S6). (D) Transcription assay shows that A-rich-$DNA_T$ (S2) but not T-rich-$DNA_T$ (S7) induces Pol V transcription stopping, and the A-rich-$DNA_T$-induced Pol V stopping has a dosage effect that relies on a sufficient number of As in $DNA_T$ (S8 and S9). All in vitro assays were performed with at least three biological repeats, with similar results. Source data are available online for this figure.

poly(U) transcripts (Fig. 4C–K). We named these complexes according to the number of continuous Us at the 3'-end of the transcripts, i.e., the 2U complex indicates two continuous Us at the 3'-end of the transcripts to form two nascent rU:dA pairs. In the 1U complex, although the major portion of RNA-$DNA_T$ hybrid, downstream and upstream dsDNAs, and $DNA_{NT}$-UPR resemble the elongation conformation with clear density that indicates a fixed conformation (Fig. 4D), the $DNA_T$ around the active site and the nascent rU:dA pair cannot be observed in the electron density (arrow in Fig. 4D), suggesting adoption of a flexible conformation. The minor conformation loss of the transcription bubble in the 1U complex implies that the A-rich-$DNA_T$-induced Pol V termination is initiated by disassembly of the first nascent rU:dA pair of the transcription bubble from the active site. Upon transcription of more As in $DNA_T$, the RNA-$DNA_T$ hybrid, mainly the newly synthesized rU:dA pairs, becomes invisible with a sequential order from downstream to upstream along with the template As feeding in, which mimics the gradual loss of the bubble conformation during transcription over A-rich-$DNA_T$ (1U to 6U complexes,

Fig. 4D–I). The coming downstream dsDNA and the connecting region between downstream dsDNA and $DNA_{NT}$-UPR gradually lose electron density and become partially disordered (arrow in Fig. 4E), suggesting a progressive transcription bubble disassembly process to release the RNA transcripts (Fig. 4C–I). These stepwise disordered transcription bubbles likely represent the bubble collapsing process, which enables the dynamics of the $DNA_T$-RNA hybrid along the nascent rU:dA pairs to block the Pol V elongation and subsequently to release the RNA transcripts, plausibly leading to a spontaneous transcription termination. Consistently, the upstream dsDNAs and the $DNA_T$-TR connected downstream dsDNA regions gradually become disordered in the cryo-EM map in the 2U and following complexes (Fig. 4E–I). The RNA-$DNA_T$ duplex is relatively rigid, while the $ssDNA_T$ after RNA release is much more flexible. Therefore, along with the template As feeding into the bubble, the RNA-$DNA_T$ hybrid gradually releases the RNA to only retain the more flexible $ssDNA_T$, which further pulls the closely connected upstream and downstream dsDNA regions to increase their flexibility.

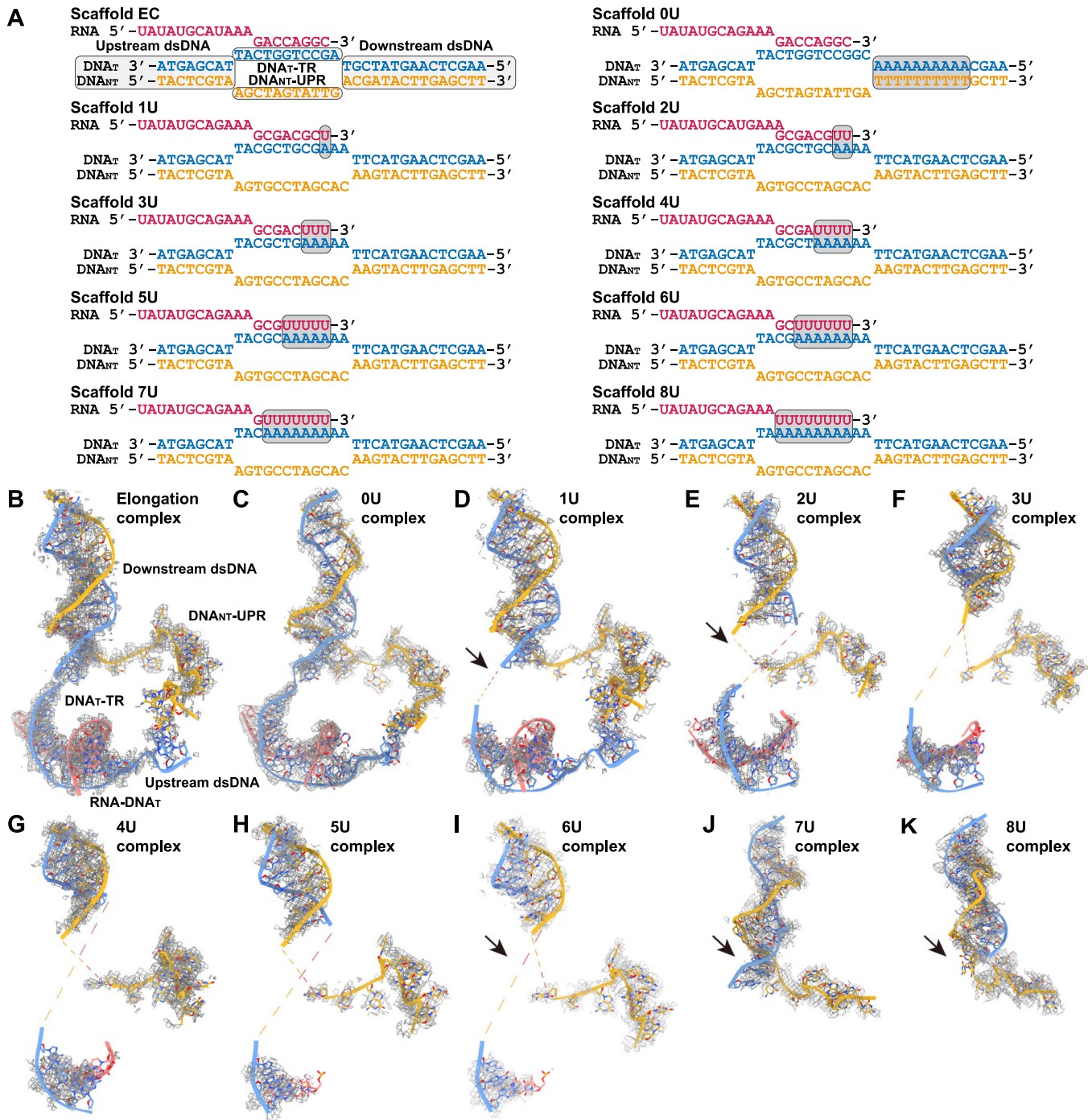

**Figure 4. Structural basis of Pol V termination.**

(A) Designed scaffolds to trap Pol V structures during transcription over A-rich-DNA$_T$. The nascent rU:dA pairs are highlighted by gray backgrounds. The segments of the designed scaffold are marked and labeled. DNA$_T$-TR, transcribing region of the template DNA; DNA$_{NT}$-UPR, unpaired region of the non-template DNA. (B) The transcription bubble conformation in Pol V elongation state (PDB code: 8HIM) was shown as a reference with the electron density map overlaid. Key structural elements of the transcription bubble are marked. (C–K) The transcription bubble conformations upon Pol V transcribing over A-rich-DNA$_T$. The transcription bubbles and the corresponding electron density maps are overlaid to show the conformational changes.

Next, the nascent RNA-DNA$_T$ hybrid in the 7U complex totally loses its density and cannot be observed (Fig. 4J). In contrast to the 6U complex, whose downstream paired dsDNA is disordered near the dsDNA branching site, the equivalent region in the 7U complex restores to clear density and connects to the DNA$_{NT}$-UPR again (arrows in Fig. 4I,J), plausibly indicating a re-annealing between DNA$_{NT}$ and DNA$_T$. Therefore, we consider that this complex plausibly represents an RNA transcript fully released conformation,

which in turn releases the $DNA_T$ to allow it to freely anneal back with $DNA_{NT}$. Because we designed unpaired sequences of $DNA_T$-TR and $DNA_{NT}$-UPR for bubble formation, we can only observe a short dsDNA fragment formation at the downstream paired region, but not a full annealing of the two DNA strands. In support of our hypothesis, the structure of the 8U complex closely resembles the 7U complex, with the downstream dsDNA and it linked $DNA_{NT}$-UPR visible but RNA and $DNA_T$-TR totally invisible (Fig. 4K). Although the transcription bubble undergoes dramatic change upon transcribing A-rich-$DNA_T$, Pol V keeps its conformation largely unchanged throughout the process (Appendix Fig. S14), suggesting that A-rich-$DNA_T$-induced Pol V termination is driven solely by the intrinsic nature of nucleic acid, implying a pure spontaneous and not cofactor-assisted termination mechanism. Moreover, the $DNA_{NT}$-UPR in all these complexes is visible with clear electron density, consistent with our previous observation of the stable interactions between $DNA_{NT}$-UPR and Pol V second subunit (Xie et al, 2023).

Mechanistically, the rU:dA pair is naturally less stable than the corresponding rA:dT and dT:dA pairs (Huang et al, 2010; Martin and Tinoco, 1980). Our structural studies showed that transcribing over A-rich-$DNA_T$ accumulates unstable rU:dA pairs, leading to disruption of the transcription bubble. Therefore, we propose an A-rich-$DNA_T$-induced spontaneous Pol V termination model. Once Pol V transcribes over a single A in $DNA_T$, slight instability arises in the transcription bubble at the nascent rU:dA pair. When a nucleotide other than A follows in $DNA_T$, the newly synthesized stable pairs can rescue the unstable transcription bubble, resulting in resumed transcription elongation and representing the state of regular transcription over discrete As. Upon transcription of continuous or dense A-rich-$DNA_T$, the accumulation of multiple unstable rU:dA pairs decreases binding between $DNA_T$ and the RNA transcript, destabilizing the RNA-$DNA_T$ hybrid from downstream to upstream. The unstable rU:dA pairs may obstruct elongation and lead to Pol V arrest, probably providing time for poly(rU:dA) pair elimination and releasing the RNA and the corresponding $DNA_T$. Finally, the released $DNA_T$ could search and reanneal to $DNA_{NT}$, therefore being released from Pol V to potentially yield a full termination.

## Discussion

Multiple transcription termination signals have been observed for the diverse RNAPs: Pols I and II can terminate at T-rich-$DNA_{NT}$ with the help of torpedo factors Rat1-Rai1; Pol III termination depends on its specific interaction with T-rich-$DNA_{NT}$; prokaryotic RNAPs can terminate at a hairpin structure preceding a poly(U) transcript (Cozzarelli et al, 1983; El Hage et al, 2008; Girbig et al, 2022b; Han et al, 2023; Hou et al, 2021; Jaiswal et al, 2016; Lang and Reeder, 1995; Reines et al, 1987). Apparently, they all transcribe over A-rich-$DNA_T$-containing DNA during termination like Pol V reported here, implying a common A-rich-$DNA_T$-based termination mechanism. Our studies suggest that the A-rich-$DNA_T$-dependent spontaneous termination of Pol V relies solely on the intrinsic nature of the transcription bubble. Consequently, the mechanism of Pol V spontaneous termination should also contribute, at least in part, to other RNAPs as a general principle.

Compared with other canonical RNAPs, Pol V features a slow elongation rate, transcription stalling, and enhanced backtracking (Haag et al, 2012; Marasco et al, 2017; Xie et al, 2023), which can yield longer retention and lead to accumulation of the unstable rU:dA hybrid upon transcribing on A-rich-$DNA_T$. This, in turn, plausibly provides sufficient time for the poly(rU:dA) pairs to fluctuate to disrupt the transcription bubble for spontaneous termination. We hypothesize that A-rich-$DNA_T$-induced spontaneous termination represents an ancestral termination pathway, which is sufficient to terminate RNAPs that elongate slowly, as observed for Pol V in the current study. However, other RNAPs, such as Pol II, transcribe faster to be able to bypass this A-rich-$DNA_T$-induced termination, as observed when Pol II transcribes through introns (Mimoso and Adelman, 2023). However, once slowed down, Pol II may also arrest on A-rich-$DNA_T$ to induce termination as in Pol V. In agreement with this hypothesis, A-rich-$DNA_T$-induced (or T-rich-$DNA_{NT}$) Pol II termination is suppressed by the elongation factors Spt4-Spt5 but stimulated by the torpedo factors Rat1-Rai1 exonuclease (Han et al, 2023). Spt4-Spt5 stimulate Pol II elongation to overcome A-rich-$DNA_T$-induced arrest and termination, protecting transcription over internal A-rich genic regions. After passing through the polyadenylation signal, Pol II releases the elongation factors and slows down (Cortazar et al, 2019), which may enable A-rich-$DNA_T$ to arrest Pol II and disrupt the transcription bubble in a Pol V-like manner (Han et al, 2023). This, in turn, may facilitate spontaneous termination of Pol II and allow time for Rat1-Rai1 to catch up with Pol II for more powerful torpedo-based termination (Han et al, 2023; Zeng et al, 2024). Consistently, purified Rat1-Rai1 is less efficient in terminating Pol II at regular sequences in vitro but accelerates Pol II termination at T-rich-$DNA_{NT}$ (Dengl and Cramer, 2009; Han et al, 2023). Therefore, A-rich-$DNA_T$-induced spontaneous termination and Rat1-Rai1 torpedo may complement each other to yield highly efficient and stepwise termination for Pol II. To confirm this, we analyzed published NET-seq data of a known Pol II *rpb1-N488D* yeast mutant, which slows down Pol II elongation (Hazelbaker et al, 2013; Topal et al, 2019). Compared with wild-type Pol II, the slow Pol II (*rpb1-N488D*) exhibits a clear termination signal at the U-track sequence of the transcripts, while the termination signals show no significant difference in G-, A-, and C-tracks (Appendix Fig. S15). These data support that A-rich-$DNA_T$-induced termination or transcription pausing may also function in Pol II (Han et al, 2023), especially when its transcription rate slows down.

Overall, we showed that the A-rich-$DNA_T$ plays a critical role in the termination of the Pol V in vitro. Consistently, the Pol V transcripts have an enrichment of the poly(U) sequence at the 3'-end, and many RdDM loci have T-rich boundaries. However, there are still Pol V transcripts and RdDM loci that do not obey these rules, suggesting other mechanisms also contribute to the termination of Pol V and the shaping of the RdDM boundary. It is noteworthy that although our NET-seq data do conform to the above model, that is, Pol V accumulates at the transcript 3'-end followed by a sharp signal decline, NET-seq is not a termination-specific assay. Methods that robustly detect in vivo RNAP termination would be needed in the future. Nevertheless, the sequence–mediated termination likely reflects only one aspect of the transcription termination mechanism of Pol V. Other mechanisms may also contribute to Pol V termination. Further

studies are required to fully reveal the Pol V termination mechanism, which defines the RdDM landscape in plants.

# Methods

### Reagents and tools table

| Reagent/resource | Reference or source | Identifier or catalog number |
|---|---|---|
| **Experimental models** | | |
| Cauliflower (*Brassica oleracea* var. *botrytis*) | This study | N/A |
| **Antibodies** | | |
| 5D2D8 | ABclonal | Custom raised |
| **Oligonucleotides and other sequence-based reagents** | | |
| DNA and RNA oligos for structural studies | This study | Appendix Table S1 |
| DNA and RNA oligos for enzymatic assay | This study | Appendix Table S1 |
| **Chemicals, enzymes, and other reagents** | | |
| Protein G agarose resin | Yeasen | 36405ES |
| 5D2D8 epitope peptide | GL Biochem | Custom synthesized |
| ATP | MedChemExpress | HY-B2176 |
| GTP | MedChemExpress | HY-113225 |
| UTP | MedChemExpress | HY-107372 |
| CTP | MedChemExpress | HY-125818 |
| Protease inhibitor cocktail | MedChemExpress | HY-K0011 |
| Micrococcal Nuclease | Beyotime | D7201S |
| T4 PNK | Thermo Fisher Scientific | EK0031 |
| TRIzol | Invitrogen | 99938601 |
| NEXTflex Small RNA-Seq Kit v3 | Bioo Scientific | 5132-06 |
| VAHTS DNA Clean Beads | Vazyme | N411-02 |
| RNase inhibitor | MedChemExpress | HY-K1033 |
| Yeast tRNA | SALMART | SEM-CR9523 |
| Next Ultra II Directional RNA Library Prep Kit | New England Biolabs | E7760L |
| **Software** | | |
| Fastp | Chen et al, 2018 | N/A |
| Hisat2 | Kim et al, 2019 | N/A |
| Samtools | Danecek et al, 2021 | N/A |
| Gencore | Chen et al, 2019 | N/A |
| Deeptools | Ramirez et al, 2016 | N/A |
| Stringtie | Shumate et al, 2022 | N/A |
| Homer2 | Heinz et al, 2010 | N/A |
| Bowtie2 | Langmead and Salzberg, 2012 | N/A |
| MACS2 | Zhang et al, 2008 | N/A |

| Reagent/resource | Reference or source | Identifier or catalog number |
|---|---|---|
| Bedtools | Quinlan and Hall, 2010 | N/A |
| T-rich-DNA_NT-analysis | https://github.com/YuxingZhou641/T-rich-DNA_NT-analysis | N/A |
| EPU | Thermo Fisher Scientific | N/A |
| Relion-3.1 | Zivanov et al, 2018 | N/A |
| MotionCor2 | Zheng et al, 2017 | N/A |
| CTFfind | Rohou and Grigorieff, 2015 | N/A |
| Coot | Emsley and Cowtan, 2004 | N/A |
| Phenix | Afonine et al, 2018 | N/A |
| PyMOL | Schrödinger | N/A |
| Chimera | Pettersen et al, 2004 | N/A |
| 3DFSC Processing Server | Tan et al, 2017 | N/A |
| **Other** | | |
| Nova Seq 6000 | Illumina | PE150 |

## Protein purification and in vitro activity assay

The DNA and RNA oligos were ordered from Sangon Biotech and GenScript, respectively, and are listed in Appendix Table S1. Cauliflower (*Brassica oleracea* var. *botrytis*) Pol V purification and the in vitro transcription assay were performed as previously described (Xie et al, 2023). In brief, the cauliflower Pol V was captured by the antibody 5D2D8 (ABclonal) from the cauliflower nucleus extraction. The antibody 5D2D8 and it captured Pol V were further immobilized on the Protein G agarose resin (Yeasen, 36405ES). After washing, Pol V was eluted by the epitope peptide (GL Biochem) as described (Xie et al, 2023). The DNA and 5'-FAM labeled RNA oligos used for in vitro assay were dissolved in a buffer of 50 mM NaCl and 20 mM Tris-HCl, pH 8.0, at a final concentration of 40 $\mu$M and 20 $\mu$M, respectively. The $DNA_T$ and $DNA_{NT}$ were incubated with a molar ratio of 1:1 and annealed by heating to 95 °C followed by gradient cooling down. The tripartite scaffold was obtained by adding the RNA to the pre-annealed dsDNA with a molar ratio of 1.1:1, heating to 45 °C for 5 min, and cooling to room temperature. The transcription assay was performed at 25 °C. The Pol V and the tripartite scaffold with a final concentration about 120 nM and 200 nM were mixed in the reaction buffer of 50 mM NaCl, 5 mM $MgCl_2$, 2 mM DTT, 10% glycerol, 0.5 mM NTPs, and 40 mM Tris-Cl, pH 8.0. After 2 h reaction, RNA loading (98% formamide, 1 mM EDTA, bromophenol blue, and xylene cyanole) buffer was added into the reaction system, and the mixture was heated to 95 °C for 5 min to stop the reaction. The reaction products were visualized by 20% denatured PAGE with 8 M urea and scanned using the blue fluorescence mode of a Tanon 6100C imager.

## Pol V NET-seq and sequencing library construction

Cauliflower Pol V NET-seq was performed as previously described with modifications (Nojima et al, 2015; Zhou et al, 2023) (Appendix

Fig. S1). To purify Pol V RNA for NET-seq, about 15 g crude cauliflower nucleus was suspended in 15 ml buffer of 25 mM NaCl, 5 mM MgCl$_2$, 2 mM DTT, 5 mM CaCl$_2$, 25% glycerol, 0.2 mM PMSF, 1× protease inhibitor cocktail (MedChemExpress, HY-K0011), and 50 mM Tris-HCl, pH 7.5. The nucleus was disrupted by osmotic shock. Micrococcal Nuclease (MNase, Beyotime, D7201S) was added into the nucleus extraction at a final concentration of 20 U/ml to digest the chromatin. After 5 min shaking at 37 °C and 220 rpm, 20 mM EDTA was added to inactivate the MNase and the digested chromatin was subjected to sonication to thoroughly release the Pol V and its bound RNA. After centrifugation at 38,000× *g* at 4 °C for 15 min, the supernatant was collected and dissolved into a buffer of 150 mM NaCl, 5 mM MgCl$_2$, 2 mM DTT, 10% glycerol, 0.05% NP40, 10 μM ZnCl$_2$, 1× protease inhibitor cocktail, and 50 mM Tris-HCl, pH 7.5. Pol V was immunoprecipitated with 100 μg 5D2D8 antibody (ABclonal) and 1 ml Protein G agarose resin (Yeasen, 36405ES) for 3 h. The resin was further washed with 100 ml washing buffer of 150 mM NaCl, 5 mM MgCl$_2$, 2 mM DTT, 10% glycerol, 0.2% NP40, 10 μM ZnCl$_2$, 1× protease inhibitor cocktail, and 50 mM Tris-HCl, pH 7.5, to remove the nonspecific binding proteins. The remaining resin-bounded Pol V RNA was used for RNA-seq library construction. The resulting RNA bound by Pol V was treated with T4 PNK (Thermo Fisher Scientific) on the resin for 7 min at 37 °C for RNA 5'-end phosphorylation. The resulting RNA was isolated by using TRIzol. RNA was precipitated by adding 1/10 volumes of 5 M NH$_4$Ac and 3 volumes of 100% ethanol, followed by overnight incubation at −80 °C. Library construction for the RNA was performed using the NEXTflex Small RNA-Seq Kit v3 (Bioo Scientific). The sequencing library was purified with VAHTS DNA Clean Beads (Vazyme) and then used for pair-end Illumina (Nova Seq 6000, PE150) sequencing. A summary of the sequencing data is presented in Appendix Fig. S2A.

## Pol V RIP-seq and sequencing library construction

To purify RNA for RIP-seq, Pol V was immunoprecipitated using a similar protocol as above but with two modifications: first, the sonication step was skipped; second, 0.4 U/μl RNase inhibitor (MedChemExpress, HY-K1033) and 200 ng/μl yeast tRNA (SAL-MART, SEM-CR9523) were added in all the buffers during purification to keep the intactness of Pol V-bound RNA. Sequencing library construction was performed as previously described (Koster and Staiger, 2021) (Appendix Fig. S1). Pol V binding RNA was isolated by using TRIzol. RNA was precipitated by adding 1/10 volumes of 5 M NH$_4$Ac and 3 volumes of 100% ethanol, followed by overnight incubation at −80 °C. The resulting RNA was used to construct the strand-specific sequencing library using the NEBNext Ultra II Directional RNA Library Prep Kit (NEB). The library was sent for Illumina sequencing (Nova Seq 6000, PE150). A summary of the sequencing data is presented in Appendix Fig. S2A.

## Pol V NET-seq and RIP-seq data analysis

For the NET-seq data, the adapter from raw reads was trimmed using fastp (version 0.20.0) with the parameter "-U -correction -overrepresentation_analysis" (Chen et al, 2018). Clean reads were mapped to *Brassica oleracea* reference genome using hisat2 (version 2.2.1) with default parameters (Kim et al, 2019). Then the non-uniquely mapped read was filtered out using samtools (version 1.9) (Danecek et al, 2021). PCR duplications were filtered by gencore

(version 0.16.0) according to unique molecular identifiers (UMIs) (Chen et al, 2019). Read 2 (R2) of the deduplicated read pair with skipped region length over 500-bp was discarded before being trimmed to keep only the 5′-nucleotide using custom scripts. For producing genome browser tracks, trimmed R2 reads were converted to a bigwig file using bamCoverage in deeptools (version 3.5.2) (Ramirez et al, 2016). For metagene profiling, RPKM values were calculated using 5-bp sliding windows, and profiles were visualized using plotHeatmap in deeptools (Ramirez et al, 2016). For RIP-seq data, the adapter from raw reads was trimmed using the fastp with default parameters (Chen et al, 2018). Clean reads were mapped to *Brassica oleracea* reference genome using hisat2 with default parameters (Kim et al, 2019; Parkin et al, 2014). The filter of the non-uniquely mapped reads and the reads with skipped region length over 500-bp were conducted as described above. R2 reads of NET-seq and RIP-seq were extracted and merged from replicates for transcripts assembly and TPM (Transcripts per million) calculation using stringtie (version 2.2.1) (Shumate et al, 2022). NET-seq transcripts were assembled with the parameter "-m 30 --fr", of which only the transcripts with TPM > 5 and length <5000-bp were retained for further analysis. RIP-seq transcripts were assembled with the parameter "-m 30 --rf" according to the library construction method, of which only the transcripts with TPM > 1 and length <5000-bp were retained for further analysis. RNA motif that is significantly enriched within −20-nt to +20-nt of the assembled transcripts end was identified by findMotifsGenome.pl with parameter "-len 8,10 -rna" in Homer2 (version 5.1) (Heinz et al, 2010). To determine the frequency of T/A/G/C tracks at 3'-end and gene body regions of Pol V transcripts, 40-bp non-template strand sequences centered on 3'-end sites or random sites at gene body regions of NET-seq transcripts are extracted to search for consecutive nucleotides. For a certain sequence, the number of non-overlapping consecutive nucleotides is counted, and the frequency is calculated as the number of motifs/the number of 40 bp regions.

## Pol II NET-seq data analysis

Pol II NET-seq data of *S. pombe* were obtained from GSE125843 (Topal et al, 2019) and processed as previously described (Nojima et al, 2015; Zhou et al, 2023). To calculate the Poly(T/A/G/C) tracts around transcript end sites of *S. pombe* genome, a 1-Kb sequence of the non-template strand centered in the annotated polyadenylation site was extracted and analyzed for each gene. Once a base repeat (>2 mer) was found, its starting coordinate and length were recorded. The starting coordinate of the maximum number of repeats (>5 mer) in a region was then marked as the position of the Poly(T/A/G/C) tracts for metagene profiling. For metagene profiling, only the 5'-end of the uniquely mapped sequencing reads, corresponding to the 3′ end of the nascent RNA, were recorded. Read density was normalized to 100,000 reads. The Pol II distribution profiles were then calculated using computeMatrix with reference-point mode and visualized using plotHeatmap in deeptools (Ramirez et al, 2016).

## Arabidopsis Pol V ChIP-seq and RIP-seq data analysis

Arabidopsis Pol V ChIP-seq were obtained from GSE100010 (GSM2667837 and GSM2667838) (Liu et al, 2018). CHH

methylation data were obtained from GSE225480 (GSM7049197) (Harris et al, 2024). Reads were then aligned to the TAIR10 reference genome with bowtie2 (v2.2.5) (Langmead and Salzberg, 2012), allowing only uniquely mapping reads with zero mismatch. Duplicated reads were removed by samtools (Danecek et al, 2021). ChIP-seq peaks were called by MACS2 (v2.2.9.1) (Zhang et al, 2008) with default parameters, and peaks with low CHH methylation level (< 0.005) were removed from analysis. Assembled RIP-seq Pol V transcripts were obtained from GSE70290 and converted into BED format for analyses (Bohmdorfer et al, 2016). Shuffled regions were generated using the shuffle function in bedtools (v2.30.0) (Quinlan and Hall, 2010). Metaplots for ChIP-seq signal and CHH methylation signal over RdDM region were generated using deeptools (v3.5.3) (Ramirez et al, 2016).

### T-rich-DNA$_{NT}$ analysis

T-scores were calculated using the TAIR10 genomic sequence. For a given position $k$ in the genome, the T-score was calculated based on the base composition of the sequence from position $k$-3 to $k + 3$ (Appendix Fig. S3B). Over this 7-bp region, the score was calculated as the number of Ts within the region, plus a bonus score for consecutive Ts, and a penalty for consecutive non-Ts. If the score was negative, it was set to zero. Specifically, given a position $k$ in the genome, $t$ equal to the number of Ts in the region of $k$-3 to $k + 3$, $c$ equal to the number of runs of consecutive Ts, $x_i$ as the length of the $i$th run of consecutive Ts in the region of $k$-3 to $k + 3$, and $g$ equal to the longest run of consecutive non-Ts in the region of $k$-3 to $k + 3$, the score was calculated as:

$$T_{Score} = \max\left(t + \left(\sum_{i=1}^{c}\sum_{j=1}^{x_i-1}j\right) - g, 0\right)$$

Examples (V = A, C or G):
```
VVTVVVV = 1 + 0 - 4 = -3 -> adjusted to 0
TTTTTTT = 7 + (1 + 2 + 3 + 4 + 5 + 6) = 28
TTVVTTT = 5 + [(1) + (1 + 2)] - 2 = 7
VTTTVVT = 4 + [(1 + 2) + 0] - 2 = 5.
```
Similarly, the AT-score was calculated using the same formula as above, but considering both As and Ts equally. For example (S = C or G):
```
SSTSSSS = 1 + 0 - 4 = -3 -> adjusted to 0
ATATATA = 7 + (1 + 2 + 3 + 4 + 5 + 6) = 28
TASSTAA = 5 + [(1) + (1 + 2)] - 2 = 7
STAASST = 4 + [(1 + 2) + 0] - 2 = 5.
```
Source code used to calculate T-score and AT-score is available on GitHub (https://github.com/YuxingZhou641/T-rich-DNA_NT-analysis). Metaplots for T-score and AT-score over NRPE1 ChIP-seq peaks and Pol V RIP-seq transcription end sites were generated using deeptools (v3.5.3) (Ramirez et al, 2016).

### Cryo-EM sample preparation

The cryo-EM sample specimens were prepared using the same protocol as previously described (Xie et al, 2023). In brief, the purified Pol V was incubated with the corresponding transcription bubble with a molar ratio of 1:2 at room temperature in a reaction buffer of 150 mM KCl, 5 mM MgCl$_2$, 2 mM DTT, 10% glycerol, 0.5 mM ATP, 0.5 mM UMPPNP, and 20 mM HEPES, pH 7.8. The complexes were purified using a Superose 6 Increase 3.2/300

column (Cytiva) and concentrated to about 0.1 mg/ml. The 300-mesh Cu R1.2/1.3 grids with 2 nm carbon (Quantifoil, Micro Tools GmbH) were pre-glow discharged by a plasma cleaner (PDC-32G, Harrick Plasma). About 4 μL of each sample was applied to the pre-glow discharged grids and blotted by a Vitrobot instrument (Thermo Fisher Scientific) with a condition of 1.5–2.5 s blotting, 5 s waiting, 6 °C temperature, and 100% humidity. The specimens were then plunge-frozen in liquid ethane pre-cooled by liquid nitrogen, and stored in liquid nitrogen.

### Data collection and processing

The cryo-EM data of the 0U, 7U, and 8U complexes were collected using a 300 kV Titan Krios G3i electron microscope (FEI/Thermo Fisher Scientific) equipped with a K3 Summit direct electron detector (Gatan) in SUSTech Cryo-EM Center with automated data acquisition executed by the software EPU under super-resolution counting mode. The data of 1U, 2U, 3U, 4U, 5U, and 6U complexes were collected on a 300 kV Titan Krios G4 electron microscope (FEI/Thermo Fisher Scientific) with a K3 direct electron detection camera (Gatan) with BioContinuum Imaging Filter (Gatan, slit width 20 eV) in PKU-IAAS Cryo-EM center with automated data acquisition by the software EPU. All data were collected at a magnification of ×81,000 (1.095 Å /physical pixel) at 20 e$^-$/Å$^2$ per second for a total dose of 50 e$^-$/Å$^2$ which fractionated into 32 frames, and the defocus values ranging from −1.0 to −2.5 μm. The cryo-EM data were processed in Relion-3.1 (Zivanov et al, 2018). Firstly, the movies were whole-frame aligned using MotionCor2 (Zheng et al, 2017), and the binning factor was set to 2 to recover the pixel size back to 1.095 Å. The data collected at PKU-IAAS Cryo-EM center were non-gain normalized, so the gain reference and upside-down flip settings were added during the motion correction process. The contrast transfer function (CTF) parameters were estimated by CTFfind v4.1 in Relion (Rohou and Grigorieff, 2015). The data processing details are shown in Appendix Figs. S5–S13.

Particles of the 0U complex dataset were template-picked with 2,466,648 particles selected among 2,808 micrographs. To accelerate the calculation, the pixel size of first-time extracted particles was expanded to 5.475 Å with the box size of 60 × 60 pixels. After two rounds of 2D classification, 1,526,121 particles were selected, and the box size was expanded to 100 × 100 pixels. Subsequently, particles with obviously low integrity were excluded by one round 3D classification, and the pixel size of the selected particles was recovered to 1.095 Å. Five more rounds of 3D classification were performed, and the remaining 77,772 particles were subjected to a consensus 3D auto-refine job, yielding a refined resolution of 3.1 Å. Finally, three rounds of CTF refinement were applied to push the refined resolution to 2.96 Å. The 1U, 5U, 6U, and 7U datasets were processed following a similar flow as 0U. In brief, 79,987, 59,869, 70,717, and 62,139 particles were selected among 4912, 3282, 6505, and 2566 micrographs for the 1U, 5U, 6U, and 7U complex datasets, respectively. After rounds of CTF refinement, the auto-refine resolutions with postprocessing were pushed to 3.46, 3.19, 3.04, and 3.42 Å, respectively. In addition to the common refinement procedures, two more rounds of Bayesian polishing were performed for the 2U and 4U complex datasets. Finally, 46,670 and 39,800 particles were selected for the 2U and 4U complex datasets to push the final resolution to 3.04 and 3.32 Å, respectively.

The 3U and 8U complex datasets were yielded by combining particles from two separate sets of data collections. Briefly, 7645

and 3995 movies were respectively collected for the 3U complex. After several rounds of 2D and 3D classifications, 34,161 and 33,879 particles were picked from the two datasets, respectively, and combined together. After two more rounds of 3D classification and CTF refinement, 28,375 particles were left to push the final resolution to 3.73 Å. For 8U dataset, 9485 and 6132 initial micrographs were respectively selected from two collections. After re-extracting the particles from the two datasets and recovering the particle pixel size to 1.095 Å, all selected particles were combined together. The final 3D auto-refined resolution can yield a 4.06 Å resolution after rounds of 3D classification and CTF refinement.

## Structure determination

The manual model building was carried out in the program Coot (Emsley and Cowtan, 2004). The structure refinement was conducted using the program Phenix (Afonine et al, 2018). A list of the parameters, including the cryo-EM data collection, structure refinement, and validation, is given in Appendix Table S2. The graphics were produced using PyMOL (Schrödinger) and Chimera (Pettersen et al, 2004). The global and directional FSC of the 3D auto-refinement were calculated using the 3DFSC Processing Server (Tan et al, 2017).

# Data availability

The structures have been deposited in the Protein Data Bank with the accession codes: 9K11, 9K12, 9K13, 9K14, 9K15, 9K16, 9K17, 9K18, and 9K19. The cryo-EM maps have been deposited in the Electron Microscopy Data Bank with accession codes EMD-61961, EMD-61962, EMD-61963, EMD-61964, EMD-61965, EMD-61966, EMD-61967, EMD-61968, and EMD-61969. The RNA-seq data have been deposited in the National Genomics Data Center under accession code PRJCA030904. Source data are provided with this paper.

The source data of this paper are collected in the following database record: biostudies:S-SCDT-10_1038-S44318-026-00763-7.

# Peer review information

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

## Acknowledgements

We thank staff members at SUSTech Cryo-EM Center and PKU-IAAS Cryo-EM Center for data collection. This work was supported by the National Natural Science Foundation of China (32325008 and 323B2041), Guangdong Basic and Applied Basic Research Foundation (2023B1515120048), China Postdoc Foundation (2024M761288), Shenzhen Medical Research Fund (A2303024), and Shenzhen Science and Technology Program (RCJC20221008092720004, RCJC20231211085945061, RCJC20231211085924032, JCYJ20240813094610015, and ZDSYS20230626091659010). JD is an investigator of the SUSTech Institute for Biological Electron Microscopy. SEJ is an investigator of the Howard Hughes Medical Institute.

## Author contributions

**Guohui Xie**: Data curation; Formal analysis; Funding acquisition; Validation; Investigation; Visualization; Methodology. **Xuan Du**: Data curation; Formal analysis; Validation; Investigation; Visualization; Methodology. **Yifang Tan**: Data curation; Formal analysis; Validation; Investigation; Visualization. **Yuxing Zhou**: Data curation; Software; Formal analysis; Validation; Investigation; Visualization; Methodology. **Cheng Chi**: Data curation; Formal analysis; Methodology. **Sixian Zhou**: Formal analysis; Investigation. **Colette L Picard**: Software; Formal analysis; Investigation; Methodology. **Songge Chai**: Data curation; Investigation. **Lei Wu**: Data curation; Investigation. **Danling Zhu**: Data curation; Formal analysis; Investigation. **Jun Zhao**: Data curation; Methodology. **Yan Xue**: Resources; Data curation. **Sisi Li**: Data curation; Investigation; Methodology. **Steven E Jacobsen**: Supervision; Investigation; Writing—original draft; Project administration; Writing—review and editing. **Zhe Wu**: Supervision; Visualization; Writing—original draft; Project administration; Writing—review and editing. **Jiamu Du**: Supervision; Validation; Visualization; Writing—original draft; Project administration; Writing—review and editing.

Source data underlying figure panels in this paper may have individual authorship assigned. Where available, figure panel/source data authorship is listed in the following database record: biostudies:S-SCDT-10_1038-S44318-026-00763-7.

## Disclosure and competing interests statement

The authors declare no competing interests.

