## [Peer Review File · The EMBO Journal]

A spontaneous termination mechanism of RNA polymerase V shapes the DNA methylation landscape in plants

Guohui Xie, Xuan Du, Yifang Tan, Yuxing Zhou, Cheng Chi, Sixian Zhou, Colette Picard, Songge Chai, Lei Wu, Danling Zhu, Jun Zhao, Yan Xue, Sisi Li, Steven Jacobsen, Zhe Wu, and Jiamu Du

Corresponding author(s): Jiamu Du (dujm@sustech.edu.cn) , Steven Jacobsen (jacobsen@ucla.edu), Zhe Wu (wuz@sustech.edu.cn)

Review Timeline:

Submission Date:	18th Nov 25
Editorial Decision:	25th Feb 26
Revision Received:	28th Feb 26
Accepted:	11th Mar 26

Editor: Cornelius Schneider

Transaction Report:

Referee #1

Comments for the Author

This work, Xie and colleagues investigate “A spontaneous termination mechanism of RNA polymerase V shapes the DNA methylation landscape in plants”. Using genomic, biochemical, and structural approaches they assess the effects of DNA sequences on Pol V termination and conclude that transcription over A-rich DNA templates results in termination due to collapse of the transcription bubble. Furthermore, they suggest this may be a common mechanism of termination for other polymerases. While these findings are intriguing, serious concerns about the data analysis and correlative nature of many of the results reduce confidence in the main findings presented in this work.

Major comments:

1. The in vivo data supporting Pol V termination at A-rich regions is not well supported.

a. Cauliflower NET-seq and RIP-seq:

-) It is not clear the methods used are accurately identifying Pol V termination sites. The authors focus on NET-seq assembled transcripts that overlap with the 3' ends of RIP-seq assembled transcripts as a proxy for termination sites. However, this corresponds to only ~1/3 of the NET-seq assembled Pol IV transcripts (2,472/8,197). Thus, it is not clear these represent the main mode of Pol V termination. Or if they instead represent a minor mode of termination that is perhaps less efficient, which is why these transcript ends are captured by the NET-seq while efficiently terminated transcripts might be selectively depleted from the NET-seq data. This latter point also draws into question the utility of NET-seq data for assessing termination as it is not specific to transcriptional termination and rather assesses where Pol V is present regardless of its transcriptional state. Additional evidence is needed to determine where Pol V is terminating before the effects of DNA sequences on this process can be investigated.

-) The data supporting Pol V arresting at the ends of transcripts in AT-rich regions is not robust.

The authors state “Sharp Pol V peaks were observed at the 3'-end of these 2,472 NET-seq transcripts by metagene analysis and at individual loci (Fig. 1b-c), suggesting Pol V arrests at these positions. Notably, our sequence motif analysis revealed a high enrichment of poly(U) signal at the 3'-end of Pol V transcripts, indicating a Pol V termination signal of T-rich-DNANT (or A-rich-DNAT, Fig. 1c-e). Thus, we show that T-rich-DNANT is highly enriched at Pol V termination sites and correlated with Pol V arrest, implying a potential DNA sequence-dependent termination mechanism of Pol V.”

- In Fig. 1b most of the signal at the 3' ends (the "sharp Pol V peaks") are from a small number of transcripts clustered at the top of the heatmap whereas most of the other NET-seq transcripts show a more even distribution of NET-seq signal. Thus, this data does not support the claim that "Pol V arrests at these (3'end) positions".
- The sequence logo around the NET-seq transcription end sites (Fig. 1d) are not very AT-rich, its only a 10 % skew (60% AT vs 40% GC). Is this AT rich compared to the rest of the genome?
- The RNA motif from the NET-seq transcripts (Fig. 1e) shows a strong U bias. However, without information about the % of transcripts with this sequence, it remains unclear if this is a common feature of the 3' ends of Pol V transcripts. Please indicated the fraction of the 2,472 NET-seq transcripts that have this motif at their extreme 3'ends.

b. Arabidopsis ChIP-seq:

-) It is not clear how well the Pol V ChIP peaks represent RdDM targets or how well the T-score is associated with termination.

By reanalyzing published Pol V ChIP-seq data 16, 044 peaks were identified. This is a higher number than expected. The paper that originally published this ChIP identified ~1,900 sites bound by Pol IV and Pol V and another ~2,300 bound by Pol V alone. Thus, it is not clear how they identified >16,000 peaks and there is not sufficient information in the methods to judge their validity/robustness. Information should be included on what parameters were used to call Pol V peaks and these peaks should be overlapped with features specific to RdDM, including CHH methylation and siRNA levels. If a significant fraction of peaks are not associated with these RdDM hallmarks, the peak calling parameters should be refined and the downstream analyses looking at the AT-richness of the peak boundaries should be repeated with using a higher confidence set of Pol V peaks.

It is also not clear how many of the 16,044 ChIP-seq peaks overlap with 4,502 Pol V RIP-seq peaks used for the T-score analysis. Furthermore, the scale of the change in the T-score at the TES is quite low (~1.6 across most of the RdDM target and then ~2.2 at the termination site; Fig. 2c). Besides at the TES, if such changes are occurring in other locations in random positions they would be missed when the analysis is centered on the TES. Please include analysis showing how the T-score changes across Pol V transcripts in an unbiased manner. For example by scanning for regions within the RIP-seq transcripts for T-scores changes in the corresponding DNA regions of 0.25, 0.5, 1, 2, 3, 4, etc. and seeing if the biggest changes are indeed restricted/enriched at the TES. If not, then the model/conclusions should be revised.

2. The biochemical assays lack replicates, and it is not clear they are measuring

termination vs measuring defects in initiation/elongation. Thus, these data fail to support the main conclusion that “As in DNAT are a key determinant to stop Pol V transcription in vivo.”

-) for all the biochemical assays in Fig. 3, replicates should be included to demonstrate the robustness of the findings and/or reveal the levels of biological variation.

-) In Fig. 3b the authors conclude that substrate 2 (S2) “almost fully eliminates transcriptional elongation of Pol V”. However, the substrate in this experiment shows much more degradation than for the other substrates or the negative control. Thus, it is not clear if the lack of elongation is due to the sequence or due to loss of the appropriately assembled substrate. The authors also state this could “imply a failure of initiation or a possible termination”. Here it is not clear how there could be a defect in termination if there is no product produced. Please include repeats with this substrate where degradation is not an issue and clarify in the discussion of the results how information about termination can be revealed by this assay.

-) In Fig. 3c the authors conclude that the “poly(A) but not poly(T) in the DNAT stopped transcription (Fig. 3c, scaffolds 5 and 6).” However, there is very little difference between the results of these experiments. S6 adds 3nt while S5 adds 2nt. Both are less efficient than the elongation of the S1 control, although it also does not elongate well. To draw conclusions from these experiments requires replicates to see if the difference between S5 and S6 is reproducible. It is also difficult to know if the difference in the results from S5 and S6 is due to effects on Pol V termination, as suggested by the text, or if it instead affects elongation. This should be made more clear in the discussion of the results.

-) In Fig. 3d, the authors conclude that “both the TATATATA and TTTTAAAA sequences of DNAT can stop Pol V transcription (Fig. 3d, scaffolds 8 and 9), revealing a dosage effect.” However in both these cases no transcription product is made, not even the +1 and +2 products, thus there is no evidence that transcription was stopped. Rather it seems that this blocks Pol V initiation. To test the role of the sequence differences present in these constructs in termination, substrates that promote better elongation are needed.

-) Given the issues raised about the ability of the biochemical substrates selected to actually assess termination, the authors should take this into consideration when interpreting the results of their cryo-EM structures.

3. NET-seq data for Pol II should be moved to the results

-) please move the discussion of Fig. S14 to the results section and make it more clear that the NET-seq assay is not a termination assay so accumulation of signal is not necessarily indicative of termination. It could also reflect pausing, which should be taken into account when interpreting this data and concluding that “A-rich-DNAT-induced termination may also function in Pol II”.

Minor comments:

-) please move the discussion of Fig. S14 to the results section and make it more clear that the NET-seq assay is not a termination assay so accumulation of signal is not necessarily indicative of termination. It could also reflect pausing, which should be taken into account when interpreting this data and concluding that U-tracks represent a conserved mechanism for polymerase termination.

-) please label Fig. 3a like Fig.4a, including marking of the DNAT /DNANT and the DNAT-TR/DNANT-UPR and the up and downstream dsDNA.

-) A-rich DNAT and T-rich DNANT are used interchangeably, which could cause confusion and make it harder to follow the manuscript. Where possible, I suggest sticking to just one way of describing these regions to avoid confusion.

Referee #2:

Comments for the Author

In this manuscript, Xie et al. investigate the transcription termination mechanism of RNA polymerase V (Pol V) using large-scale and high-purity cauliflower-derived Pol V in combination with NET-seq and RIP-seq. The data from these experiments identify enrichment of poly(U) sequences at the termination sites of Pol V-associated transcripts. Furthermore, in vitro transcription assays using nucleic acid scaffolds designed based on these findings demonstrate that scaffold RNA fails to elongate when an A-rich DNAT strand is paired with a complementary poly(U) RNA sequence. Structural analyses by cryo-EM of Pol V complexes with a series of scaffold RNAs, each containing increasing numbers of 3'-terminal U residues, reveal that as the number of U residues increases, the transcription bubble progressively disassembles from the Pol V active center. The authors interpret this as being caused by the formation of unstable

rU:dA base pairs at the 3' end of the RNA, rather than by structural changes in Pol V itself. These structural insights support the authors' proposed model of A-rich DNAT-induced spontaneous Pol V termination. Overall, the experiments are well designed, and the resolution and statistical validation of the cryo-EM models seem sufficiently supported, although some anisotropy is observed in the cryo-EM data. The study provides strong and novel evidence addressing a previously unresolved aspect of Pol V function—its transcription termination mechanism—in the context of plant-specific epigenetic silencing. However, further discussion is warranted regarding the relationship between the number of A residues in the A-rich DNAT sequence, the observed disassembly of the transcription bubble from the Pol V active center, and the actual cessation of RNA elongation by Pol V.

Major concerns:

1. Is backtracking occurring in the Pol V–scaffold complex structures? For instance, in the 1U complex, the absence of density for the rU:dA base pair region might be related to backtracking. The authors should discuss the potential relationship between this lack of density and backtracking.

2. In the Pol V–2U scaffold complex, there is a loss of cryo-EM density not only in the region of the rU:dA base pairs and in part of the downstream dsDNA, but also in the upstream dsDNA, which is spatially and sequence-wise distant. This observation raises the question of what mechanistic explanation could account for such unexpected long-range disassembly.

3. How might disassembly of the upstream dsDNA affect transcription elongation by Pol V? While direct experimental evidence may be difficult to obtain, it would strengthen the interpretation if the authors correlated the cryo-EM structural data with *in vitro* transcription assays using the same 1U to 6U scaffold RNAs. Furthermore, in the case of Scaffold 9, transcription activity is impaired despite the presence of four uridine residues at the 5' end—not the 3' end—of the complementary RNA strand. This suggests that the observed rU:dA-induced disassembly at the RNA 3' end may not fully explain the loss of elongation activity, and additional factors or mechanisms might be involved.

4. To better understand how Pol V transcription termination influences the DNA methylation landscape, this reviewer suggests extending the *in vitro* transcription assay beyond UA pairs to also include UG and UC combinations in Scaffold 8 derivatives.

Minor concerns:

1. In the lower 2D heatmap of Fig. 1b, the signal trends are somewhat difficult to discern. Please adjust the heatmap color scale to better highlight the high signal levels observed at the ends of each gene.

2. In Figure 3b (S2), degradation is substantial, making comparison with other scaffolds difficult. Please consider replacing it with data showing uniform 20 nt bands across scaffolds.

3. Throughout the manuscript, the use of technical terminology related to RNA polymerase may hinder accessibility for the broader readership of XXX. This reviewer recommends briefly defining such specialized terms upon their first mention.

We thank the reviewers for their suggestions to help improve the manuscript. To some of the comments, we have our different understanding and we have our point-to-point response below with our response highlighted in red.

Reviewer #1 (Comments for the Author):

We thank the authors for their efforts in revising this manuscript and offer additional comments based on our previous top concerns below.

1a. While the additional information provided about the cauliflower NET-seq and RIP-seq address some of my concerns, my reservations about the limitations in this assay to assess Pol V termination and draw global conclusions remain. Especially considering the small fraction of transcripts being focused upon (2,472/8197) and the relatively subtle sequence features being equated to termination. However, the new analyses have provided additional support that at this subset of transcripts have an enrichment of U-rich motifs specifically at their 3' ends. Relating to these experiments, I have the following comments:

We thank Reviewer 1 for the insightful comments. The reason why we focused on 2,742 transcripts out of the total 8,197 NET-seq transcripts has already been explained in detail in our previous response. Briefly, this selection was made to ensure rigor by analyzing only the high-confidence transcripts that are supported by both NET-seq and RIP-seq data.

It is important to emphasize that although our conclusions were primarily based on these 2,742 transcripts, namely, (1) Pol V exhibits characteristic pausing at transcript 3' ends, suggesting transcription termination, and (2) non-template strand T-tracks frequently occur at these ends, these features are not unique to the 2,742 transcripts. In fact, they are also present, and even more pronounced, across the entire set of 8,197 transcripts (see Response Fig. 1 and Fig. 4).

Regarding the reviewer's concern about "relatively subtle sequence features being equated to termination," we respectfully disagree. We believe this concern may reflect a different expectation of how such sequence features manifest in the context of transcription termination. Considering that Pol V transcribes across tens of thousands of dispersed genomic loci, it is very unlikely that any single sequence element alone could fully explain its transcription termination. Extracting key sequence signals from such complex backgrounds and understanding the underlying "grammar" is itself a major frontier in molecular biology. Our data show that more than 25% of Pol V transcripts display four or more consecutive T tracks at their 3' ends (Fig. 1e). Among the 8,197 transcripts analyzed, over 2,000 exhibit Pol V pausing at the 3' end (Response Fig. 1, right panel). These data provide strong evidence that consecutive non-template T tracks represent the most prominent and observable sequence feature associated with Pol V transcription termination. For comparison, Pol II has a much lower probability of pausing or terminating at consecutive T tracks (PMID: 37683646, PMID: 39496457). In our analysis, this phenomenon is only observed in a mutant where transcription

elongation is slowed down (Fig. S14). Therefore, we are confident that our study reveals a distinct and defining sequence-dependent termination feature of Pol V that clearly sets it apart from Pol II.

o It remains unclear from Fig 1b (even with a color scale change) how much the 3' peak is being influenced by a few transcripts showing strong NET-seq signal at their 3' ends. I suggest clustering these transcripts into groups (K-means clustering) to see if the strongest peaks are at the 3' ends across the majority of the transcripts or if most transcripts have an even distribution of signal across their gene body.

We thank the reviewer for his/her careful and insightful examination of our results. We acknowledge that the assessment of Pol V enrichment at transcript 3' ends is partly subjective and may depend on how one visually interprets the heatmap, which may explain why the signal did not appear sufficiently prominent to the reviewer. Indeed, the intensity of signals in the heatmap is influenced by variation in expression levels among transcripts. To address this concern, we identified the position of the maximum NET-seq signal for each transcript and plotted the distribution of these peak positions relative to the transcript end. As shown in Response Figure 1, for both the subset of 2,472 transcripts and the entire set of 8,197 transcripts, a substantial proportion of NET-seq highest peak signals occur at the 3' end, corresponding to more than 2,000 transcripts out of the total 8,197 transcripts. We also attempted the K-means clustering analysis as suggested by the reviewer. However, we found that the results were heavily biased by expression levels instead of the distribution pattern, making it difficult to obtain a fair outcome. For this reason, we decided not to pursue this approach further. Nevertheless, the distribution analysis is fully consistent with our original heatmap observations (Original Fig. 1b and Fig. S2C; also see Response Fig. 2), showing that Pol V frequently pauses at transcript 3' ends in both the 2,472 and the 8,197 transcript datasets. Finally, we would like to emphasize that for Pol V, the pausing at the 3' ends is clearly and frequently visible in the raw data tracks. To make this feature more intuitive, we have provided additional representative examples (Response Fig. 3), all of which strongly support that Pol V pausing and termination at transcript ends is a robust and prominent feature, rather than a rare occurrence.

Response Figure 1. Distribution of the position of the maximum NET-seq signal along

each transcript. 0 indicates the transcription start site (TSS), and 1 indicates the transcription termination site (TTS). The left panel shows the results for the 2,472 NET-seq transcripts, and the right panel shows the results for all 8,197 transcripts.

Response Figure 2. Metagene plots showing the distribution of Pol V NET-seq signals along 2,472 transcripts (left) and 8,197 transcripts (right).

Response Figure 3. Genome browser tracks showing the normalized counts from Pol V NET-seq data (1-nt resolution) and reads from Pol V RIP-seq data of eight different example Pol V transcripts. Transcripts are indicated by black boxes. Arrows indicate the direction of transcription.

o The new data analysis assessing the occurrences of U-rich motifs at the ends of the 2,472 transcripts (Fig. 1e, Fig. S2i, and rebuttal Response-Fig 4) provide stronger evidence for an enrichment of this motif at the ends relative to the body of the transcripts. However, it is not clear why Fig. 1d presents a motif that is only present at the end of 58 of the top 1000 transcripts when the data in the response to reviewers shows a shorter U-rich motif present in over 50% of the 2,472 transcripts. Please clarify.

We thank the reviewer for acknowledging our evidence (Fig. 1e, Fig. S2i, and previous rebuttal Response-Fig 4) supporting the enrichment of the U-rich motif at transcript 3' ends. Regarding the reviewer's comment on Figure 1d, that only 58 out of 1,000 transcripts were found to contain the U-rich motif, we have already explained briefly in our previous response. First, this result was generated using a specific software, MEME, and the outcome is inevitably influenced by the algorithm, parameter settings,

and other implementation details of the software. We regret that we lack sufficient expertise in the underlying algorithmic principles to fully explain why MEME did not enrich for shorter U-rich motifs, whereas HOMER did, and we believe this issue seems falls outside the scope of our research. Based on our experience, MEME employs a global optimization strategy favoring long, high-information motifs, whereas HOMER detects shorter, high-frequency motifs. Therefore, for simple U-rich elements, MEME tends to identify a longer motif present in fewer sequences, while HOMER captures the short motif more broadly. Indeed, it is clear that MEME identified a relatively long motif (15 nt), while the input sequences were only 20 nt in length. Therefore, it is expected that only a small proportion of the sequences would contain this longer motif. Importantly, as we have shown previously through additional analyses that do not rely on any specific motif-finding software (Fig. 1e), U-rich motifs are indeed frequently present at the 3' ends of transcripts. For example, stretches of four or more consecutive Ts are observed at the 3' ends of over 25% of transcripts (Fig. 1e), supporting the robustness of this sequence feature. The motif enrichment analysis was therefore intended as an alternative way to illustrate the same underlying pattern. Moreover, both MEME and HOMER identified U-rich motifs, although with clear differences in motif length and the frequency with which the motif appears. Additionally, we also performed HOMER motif enrichment analysis for all the 8197 NET-seq transcripts, and a U-rich motif was highly enriched as well (Response Fig. 4). In light of the reviewer's comment and to avoid unnecessary confusion, we have decided to replace the MEME-derived motif enrichment result in Figure 1d with the corresponding result generated by HOMER.

Response Fig. 4 U-rich motifs were identified by HOMER in both the 2472 transcripts (Left) and the 8197 transcripts (Right) at high frequency.

1b. The additional analyses presented provide support that the regions identified as Pol V peaks (~10K of the initially reported 16K) are well correlated with RdDM features (CHH methylation) and overlap well with previously published Pol V transcripts. Furthermore, analysis of the T-score across the full Pol V transcripts shows a high T-score at both the TSS and TES but not in the body of the transcript. While the role at the TSS is unclear, these findings support the model that T-rich regions are enriched at

the ends of Pol V transcripts in Arabidopsis.

The enrichment of AT-rich sequences at the TSS sites is a common feature for almost all RNA polymerases. The initiation of transcription requires the separation of the two strands of the DNA duplex; the AT-rich sequence is easily separated. Therefore, we consider that the high T-score in TSS also indicates an initiation signal like other RNA polymerases, and we have mentioned it in our manuscript. Moreover, a high T-score in TSS is related to a separate story about the Pol V initiation that we are just working on.

2. The authors provided replicates for all the biochemical experiments as requested and added additional text to point out potential causes for the increased degradation of S2 and limits for the interpretations of the data comparing S5 vs S6 and S8 vs S9. While more transparent, these results are on their own only able to weakly support the authors conclusions, shifting additional burden of proof on the genomics and structural data.

Indeed, it is very challenging to distinguish between transcription pausing, where RNA polymerase stops moving along DNA but remains bound to it, and termination, where RNA polymerase stops moving along DNA and detaches from it, in our in vitro assay. We agree with the reviewer that the in vitro data itself is weak. That is why we performed a series of structural snapshots to directly observe the termination process at the molecular level after proposing the termination mechanism based on sequence analysis and biochemical assays. Our conclusion derives from the combination of all the data.

3. The authors' rationale for keeping Fig S14 in the discussion is reasonable and they address all my other minor comments satisfactorily.

We thank the reviewer.

Reviewer #2 (Comments for the Author):

The authors have addressed all of my comments, providing persuasive responses and making appropriate revisions to the manuscript with supplementary explanations where necessary. I find their responses satisfactory, and therefore I have no further comments.

We thank Reviewer 2 for support.

Dear Dr. Du,

Thank you for submitting your manuscript which was previously reviewed at a different venue to The EMBO Journal. Your study has now been seen by an arbitrating referee, who finds that the remaining concerns by the original referees have been addressed and recommends publication of the manuscript. There remain only a few mainly editorial points that have to be addressed before I can extend formal acceptance of the manuscript:

- Please remove the figures from the manuscript file and upload as individual files with sufficient resolution/quality for production.
- Please rename the "Data and materials availability" section to Data Availability
- Please rename the Conflict of Interest section into "Disclosure and Competing Interests Statement", in accordance with our updated Guide to Authors (<https://link.springer.com/partners/embo-press/editorial-policies#Competing%20interest%20disclosures>)
- As we are switching from a free-text author contribution statement towards a more formal statement based on Contributor Role Taxonomy (CRediT) terms, please remove the present Author Contribution section and instead specify each author's contribution(s) directly in the Author Information page of our submission system during upload of the final manuscript. See <https://casrai.org/credit/> for more information.
- Please adjust the format of the reference list and of the in-text citations according to EMBO Journal format (alphabetical order, author name et al + year.../up to 10 author names in the reference list before et al / please refer to our Guide to Authors for additional information on EMBO J reference format).
- Please provide either a "Yes" or a "Not Applicable" answer to each one of the questions in your Author Checklist (<https://media.springernature.com/original/springer-cms/rest/v1/content/27825796/data/v1>). In the last column of this checklist, only the sections of the manuscript where the relevant information can be found should be listed (the information per se should be included in the main manuscript file).
- Please double-check to make sure to all relevant funding information is entered in both the manuscript and also into our submission system: Check if SUSTech Institute for Biological Electron Microscopy and the Howard Hughes Medical Institute need to be entered as separate funders since two authors are investigators at these institutes.
- APPENDIX FILE WITH TABLE OF CONTENT: Please start with a title page that has Appendix to [Article title] and a table of contents with page numbers listing all items; the correct nomenclature should be Appendix Figure S1, etc. and Appendix Table S1, etc.; "Supplementary Data" should not be used; ms callouts need to be updated too
- Please provide suggestions for a short 'blurb' text prefacing and summing up the conceptual aspect of the study in two sentences (max. 250 characters), followed by 3-5 one-sentence 'bullet points' with brief factual statements of key results of the paper; they will form the basis of an editor-written 'Synopsis' accompanying the online version of the article. Please also provide an altered synopsis image, making sure that the aspect ratio conforms to our website's format - it should be exactly 550 pixels wide and between 300-600 pixels high.
- Please provide the Reagent and Tools Table. For more information, please check <https://media.springernature.com/original/springer-cms/rest/v1/content/27825802/data/v1>
- Please adjust the section order of the manuscript to be in the following: Title page - Abstract & Keywords - Introduction - Results - Discussion - Methods - Data Availability - Acknowledgments - Disclosure Statement & Competing Interests - References - Figure Legends - (Main Tables with legends if applicable) - Expanded View Figure Legends.
- Please provide the specific URLs for 9K11, 9K12, 9K13, 9K14, 9K15, 9K16, 9K17, 9K18, 9K19, EMD-61961, EMD-61962, EMD-61963, EMD-61964, EMD-61965, EMD-61966, EMD-61967, EMD-61968, EMD-61969, PRJCA030904 datasets in the data availability statement.

With best regards,
Cornelius Schneider

Cornelius Schneider, PhD
Editor | The EMBO Journal
c.schneider@embojournal.org

Please refer to our figure preparation guideline in order to ensure proper formatting and readability in print as well as on screen:

<https://link.springer.com/journal/44318/submission-guidelines#cms-Figure-and-data-presentation>

Use the link below to submit your revision:

Referee #2:

After another revision, the manuscript by Xie et al has been significantly improved, and the final conclusions confirming the Pol V pausing and termination at the transcript ends are now much better supported. In particular, the inclusion of NET-seq and RIP-seq data for all 8,197 transcripts, as well as replacing MEME motifs by HOMER motifs was a valid solution. I also think that the explanations and responses fully address comments and reservations of Reviewer #1.

We thank the reviewer for the positive comments and have provided a response below.

Referee #2:

After another revision, the manuscript by Xie et al has been significantly improved, and the final conclusions confirming the Pol V pausing and termination at the transcript ends are now much better supported. In particular, the inclusion of NET-seq and RIP-seq data for all 8,197 transcripts, as well as replacing MEME motifs by HOMER motifs was a valid solution. I also think that the explanations and responses fully address comments and reservations of Reviewer #1.

We thank the reviewer for the positive comments.

Dear Dr. Du,

I am pleased to inform you that your manuscript has been accepted for publication in the EMBO Journal.

You may qualify for financial assistance for your publication charges - either via a Springer Nature fully open access agreement or an EMBO initiative. Check your eligibility: <https://link.springer.com/journal/44318/how-to-publish-with-us>

Yours sincerely,

Cornelius Schneider, PhD
Editor
The EMBO Journal
c.schneider@embojournal.org

Please note that it is The EMBO Journal policy for the transcript of the editorial process (containing referee reports and your response letters) to be published as an online supplement to each paper. If you should prefer removal of any referee-only figures included in the point-by-point response(s), e.g. because they may still be used for future publication or because they have been reproduced from published work by others, please do let us know immediately via response email.

More information is available here: <https://link.springer.com/partners/embo-press/editorial-policies#Peer%20review>
